# Pooled optical screening in bacteria using chromosomally expressed barcodes
Ruben R. G. Soares [1,4], Daniela A. García-Soriano [2,4], Jimmy Larsson[2,4], David Fange [2,4] ✉, Dvir Schirman [2], Marco Grillo [1], Anna Knöppel[2], Beer Chakra Sen[2], Fabian Svahn [2], Spartak Zikrin [2], Michael Ratz [3], Mats Nilsson [1] ✉ & Johan Elf [2] ✉

Optical pooled screening is an important tool to study dynamic phenotypes for libraries of genetically engineered cells. However, the desired engineering often requires that the barcodes used for in situ genotyping are expressed from the chromosome. This has not previously been achieved in bacteria. Here we describe a method for in situ genotyping of libraries with genomic barcodes in *Escherichia coli*. The method is applied to measure the intracellular maturation time of 84 red fluorescent proteins.

Optical pooled screening is a powerful tool to connect genetic alterations to live-cell phenotypes for libraries of strains[1]. The strains in the library are phenotyped under the microscope, without knowledge of their genotype. The genetic identities of the cells are only revealed by in situ genotyping after the cells have been fixed[2–5].

In situ genotyping of bacterial cells has only been possible in cases where the strain-specific phenotypes are induced by a medium to high-copy-number plasmid, ensuring sufficient expression of the genotyping signal. This approach has been used to study the brightness of fluorescent proteins expressed from plasmids[4] and to characterise CRISPR knockdown libraries in which a sgRNA is expressed from the same plasmid as the genotyping barcode[6]. Many experimental settings do however require the expression of genotyping barcodes from the chromosome. In some experimental designs, variations in the plasmid copy number between different cells could potentially mask differences in the phenotype of interest, as for example when studying noise in gene expression. In other cases, the need for chromosomal expression might be a strict requirement of the experiment to be performed. For example, when studying libraries of fluorescently labelled chromosomal loci, the locus label would be integrated together with a barcode to enable efficient identification of the strain. Similarly, when studying the physiological effects of a mutation in a chromosomal locus, the genotyping barcode would typically be expressed near the locus itself. For the benefit of these experimental designs, we have developed a method for genotyping chromosomal barcodes in *E. coli*.

## Results and discussion

To demonstrate the use of the method, we chose to characterise the average time it takes from expression until the protein becomes fluorescent, i.e. the maturation time, of 84 fluorescent proteins (FPs) excitable using 575 nm light. The maturation kinetics of red fluorescent proteins[7] is often

problematic for bacterial applications, because red FPs often have a longer maturation time than the generation time of many bacteria. In these cases, less than half of the FP molecules will be mature at any point in time if expressed under a constitutive promoter. In addition, the slow maturation time of red FPs makes them poorly suitable for studying dynamics of fast cellular processes, as they tend to 'blur' the fluorescent readout in time.

We produced a library of bacterial strains expressing different red FPs from the chromosome. In this case, chromosomal expression normalises the expression levels across strains, and makes the brightness at the selected excitation wavelength directly comparable.

We primarily selected the fluorescent proteins from FPbase[8] and introduced the corresponding *E. coli* codon-optimised open reading frames (ORFs) replacing *lacZ*, *lacY* and *lacA* of the *lac* operon. Each ORF is associated with a unique barcode sequence that is flanked by a T7 promoter and RNA stabilising elements to increase the abundance of barcode RNA (Fig. 1A & Supplementary Fig. 1). Sequences common to all strains are tabulated in Supplementary Table 1, while the complete gene fragments used to construct each strain are tabulated in Supplementary Data 1. The strains were constructed individually (see 'Methods'), but pooled into a strain library before loading into a mother machine type polydimethylsiloxane (PDMS) microfluidic chip with 4000 cell traps[9]. Each trap is only open to cells in one end and fits a single file of 10–16 cells. This implies that after 4 generations, each cell in the trap is a direct descendant of the cell at the bottom, i.e. at the closed end and all cells from other lineages have been pushed out. Since all cells in the same trap descend from the same mother cell, they clonally express the same FP (Fig. 1B).

We initiated the maturation experiment with fully induced FP expression at 1 mM IPTG, and imaging in steady-state growth conditions for 1 h. After this, the media was swapped to media including 250 ug/ml of chloramphenicol (CHL), that within a few minutes stops protein synthesis

[1]Department of Biochemistry and Biophysics, Stockholm University, SciLifeLab Stockholm, Stockholm, Sweden. [2]Department of Cell and Molecular Biology, Uppsala University, SciLifeLab Uppsala, Uppsala, Sweden. [3]Department of Cell and Molecular Biology, Karolinska Institute, Stockholm, Sweden. [4]These authors contributed equally: Ruben R. G. Soares, Daniela A. García-Soriano, Jimmy Larsson, David Fange. ✉e-mail: david.fange@icm.uu.se; mats.nilsson@scilifelab.se; johan.elf@icm.uu.se

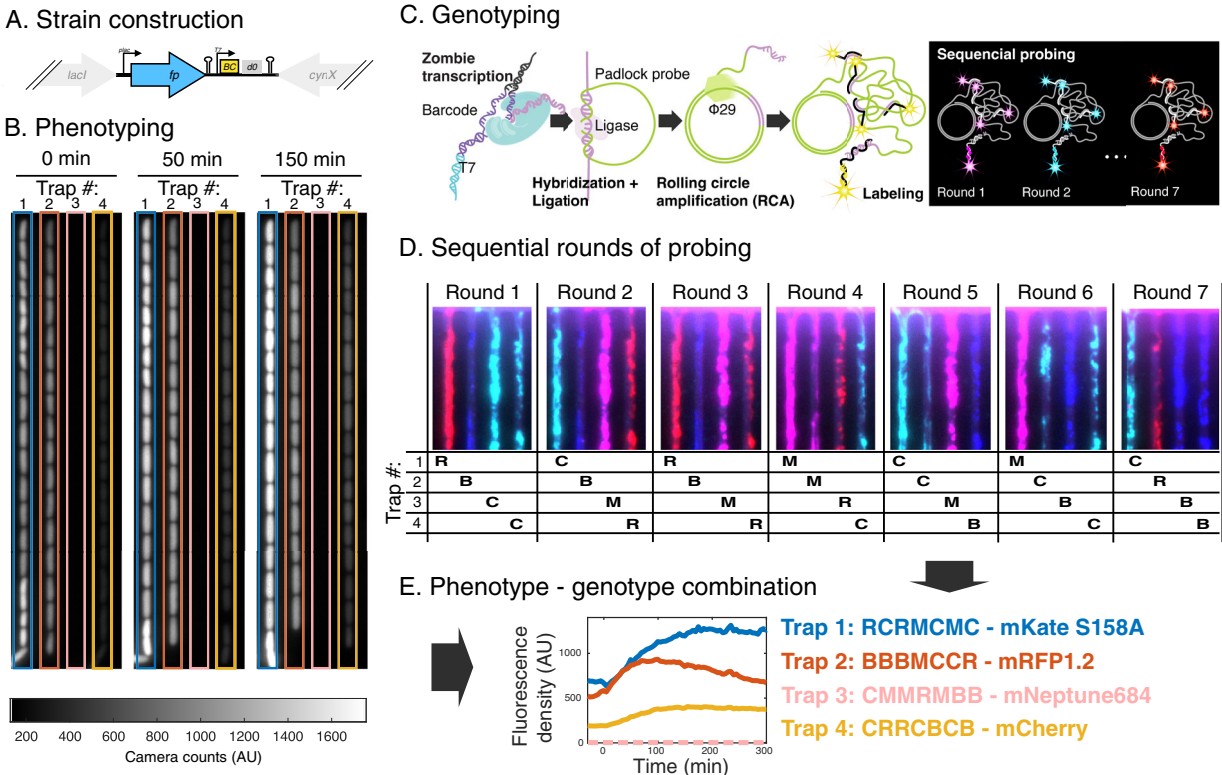

**Fig. 1 | Phenotyping followed by in situ genotyping. A** Cartoon highlighting the inserted fluorescent protein ORF (fp) and barcode (BC) in its chromosome context where it replaces the genes *lacZ*, *lacY* and *lacA*. The d0 sequence is added for mRNA stability. **B** Example of fluorescence images of cells at three different timepoints after switching to chloramphenicol-containing medium. Time point zero indicates swap to medium with chloramphenicol. **C** Cartoon showing major steps of in situ genotyping. **D** Fluorescence images of seven rounds of genotyping for the cell traps shown in (**B**). **E** Quantified fluorescence density throughout the experiment for one example lineage for the traps highlighted in (**B**) together with the decoded genotypes from (**D**). None of the cells in Trap 3 passed the filtering criteria on the cell fluorescence which is indicated by a dashed line at zero.

(Supplementary Fig. 2), which, in turn, also eventually stops cell growth (Supplementary Fig. 3). The already expressed, but still immature, FPs will become fluorescent over time and, in turn, the total cell fluorescence will increase (Fig. 1B, E). The increase of the fluorescence signal was monitored by time-lapse fluorescence microscopy, repeatedly exposing the same cells to 575 nm light every 5 min. The cell growth was monitored in phase contrast every minute. The image acquisition ran for a total of 6 h after which most cells approached a plateau in fluorescence intensity or even showed a fluorescence intensity decrease due to photobleaching.

After collecting the phenotyping data, the genotypes for each cell trap were determined in situ. The cells were fixed and permeabilized in 70% EtOH and rehydrated in PBS-Tween before the cell walls were partly degraded by lysozyme. The RNA barcodes were expressed using zombie transcription, i.e. by adding T7 polymerase to the already fixed cells (Fig. 1C)[10]. Barcode-specific ssDNA padlock probes (PLPs) were hybridised directly to the RNA barcodes, ligated by splintR-ligase[11] and amplified by rolling circle amplification (RCA)[12]. This amplification of the barcode signal, not present in our previous plasmid-based method[2,13], makes it possible to read out individual barcodes by combinatorial FISH hybridisation[14,15] to the RCA products. This enables the detection of a large number of barcodes with a limited set of differently coloured fluorescent probes[14,15]. In each sequential hybridisation round, a mix of adaptor 'L-probes' were hybridised to the RCA products, followed by parallel hybridisation with 4 differently coloured fluorescent detection probes. Combinatorial barcode readout maximally allows identification of C^N different barcodes where C is the number of colours and N is the number of rounds of probing[14,15]. (See 'Methods' for details of the genotyping protocol). To unambiguously map the phenotype to the

genotypes at the single trap level we used unique position identifiers imprinted in the microfluidic device.

For barcode readout, each of the barcodes were assigned a unique sequence of the four differently coloured detection probes, one colour for each of the seven rounds of genotyping. The minimal Hamming distance between the detection probe sequences for the different genotypes was set to 3, allowing correction of a single error in the detection probe sequence[15]. See Table 1 for details of the genotyping performances in three replicate experiments, where each replicate experiment is started from the strain library cryostock. By counting the number of single round errors among the decoded traps we assess an average error probability of 0.038 per trap per round. Assuming that decoding errors are uncorrelated between padlock sequences and rounds of probing, this error probability results in a probability of ~0.002 for assigning a cell trap with the wrong genotype. The efficiency of each barcode is estimated by comparing the number of decoded traps for each barcode (Fig. 2B) with the number of reads from NGS-based amplicon sequencing of the barcodes (Fig. 2A). Overall, we find good agreement between in situ and amplicon sequencing with 73 barcodes having less than a two-fold change in the readout frequency of the in situ sequencing as compared to what is expected from the amplicon sequencing (Fig. 2C). However, the frequency of over- or under-representation in the in situ genotyping (15%) is higher than expected from the probabilistic loading of 85 strains into a finite number of microfluidic traps, which suggests a barcode sequence dependent variability in the barcode readout efficiency.

We quantify the maturation time for each lineage by fitting a single exponential function to the fluorescence intensity integrated over the cell area after background subtraction (Methods). For the FPs where the cell fluorescence at the last point of measurement has

bleached by more than 15% compared to the peak value (FusionRed-MQV, mRFP1, mRFP1.2, mScarlet-I and mScarlet3), a function containing the sum of two exponential functions was used in order to account also the bleaching rate.

The maturation times for individual cell lineages together with the average cell-fluorescence intensity in the last three frames of fluorescence imaging before switching to chloramphenicol-containing media are presented in the scatter plots in Fig. 3. Note that the brightness is not likely to be the maximal brightness for each fluorophore since all of them are excited using the same light source and the excitation spectra for different fluorophores peak at different wavelengths. The results from the three replicate experiments are shown in different colours in Fig. 3. Single dots in the scatter plots of Fig. 3 correspond to measurements from single cell lineages. Statistics of the maturation times shown in Fig. 3 are presented in Fig. 4 and Supplementary Table 2. When the maturation time is longer than the experiment duration after the switch chloramphenicol-containing medium (300 min), the time constant is reported as >300 min, since these values could not be accurately determined. The majority of the cell lineages for each barcode in Fig. 3 fall into one major cluster. Using the phenotypic outliers we estimate that ~3% of the cell traps contain cells with a phenotype that deviate from the majority (Supplementary Table 3A). Two barcodes are however highly overrepresented with respect to phenotypic outlier frequency: pHuji in EXP-24-CB5647 (Supplementary Table 3A, Fig. 3), and the no FP control strain barcode (Supplementary Table 3B). This partly explains why the

**Table 1 | Genotyping performance in three replicate experiments (CB5644, CB5647, CB5648)**

| Exp. id | Nr. of traps with cells[1] | Traps with no signal[2] | Traps with double signal[3] | Unassigned decoded traps[4] | Successfully decoded traps[5] | Decoding error probability per round[6] |
|---|---|---|---|---|---|---|
| CB5644 | 3159 | 639 | 210 | 98 | 2212 [70%] | 0.031 |
| CB5647 | 3076 | 459 | 263 | 197 | 2157 [70.1%] | 0.052 |
| CB5648 | 3256 | 432 | 98 | 87 | 2726 [83.7%] | 0.032 |

Each experiment was performed on a microfluidic chip with 4000 traps. 1—After performing the enzymatic steps some traps were left with no permeabilized cells, thus we consider only traps that stain for DNA (DAPI) after in situ genotyping. 2—Traps that included permeabilized cells but produced no fluorescence signal during the rounds of sequential probing. 3—Traps that fluoresced in more than one colour at a given sequential probing round. 4—Traps where the read out sequence of detection probes was more than one symbol away from any barcode in the library and could thus not be error corrected. 5—number of traps that produced a valid barcode signal that was successfully assigned to a library barcode either directly or using error correction. 6—Assessment of the error probability per trap per round based on error-corrected decoded barcodes.

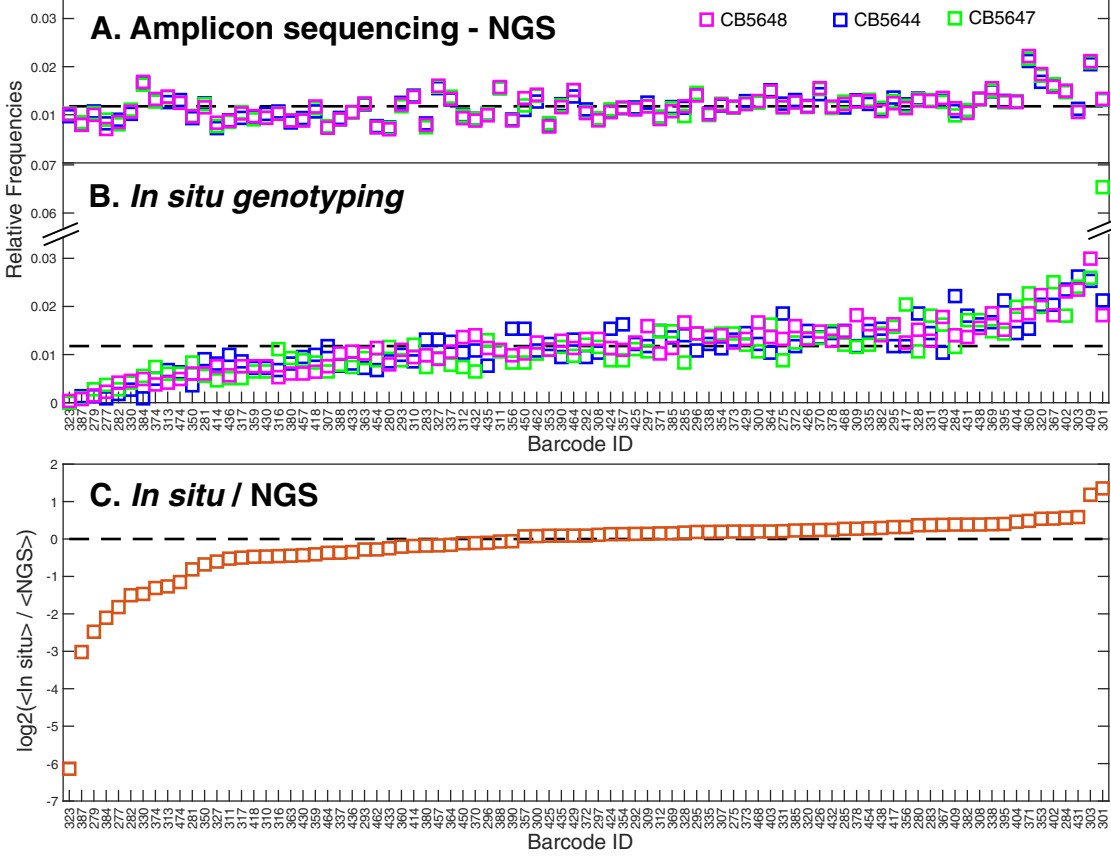

**Fig. 2 | In situ genotyping barcode efficiency.** Normalised frequency of barcodes detected by NGS-based amplicon sequencing (**A**) and in Situ genotyping (**B**). The dashed black lines indicate equal frequencies of the 85 barcodes. Different colours represent replicate experiments (blue: CB5644, magenta: CB5648 and green: CB5647). **C** The averages of the three experiments in (**B**, <in situ>) divided by the average of the three experiments in (**A**, <NGS>). The black dashed line indicates <in situ>/<NGS> = 1.

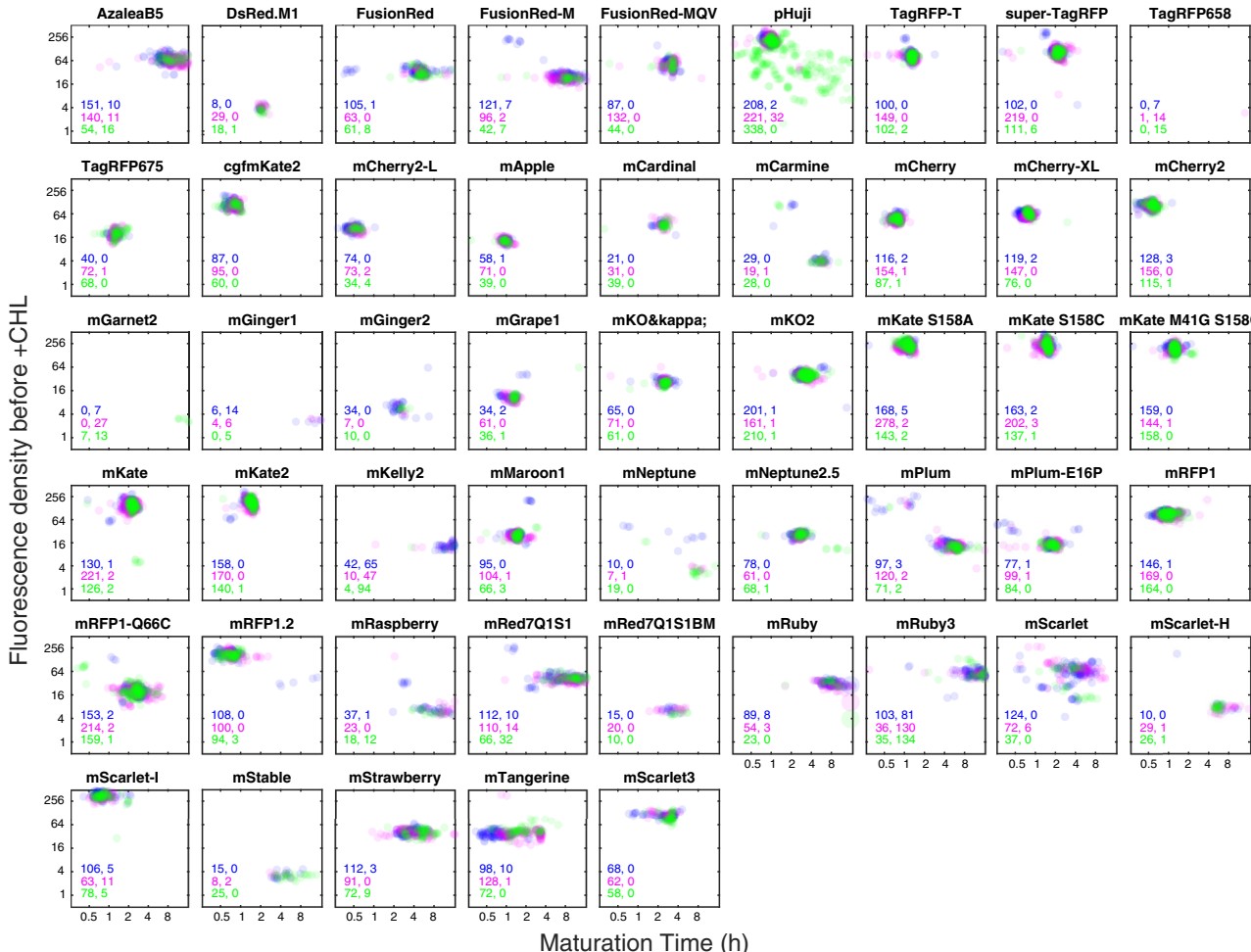

**Fig. 3 | Scatter plots for the maturation time (x-axis) and average cell fluorescence before + CHL (y-axis) for the different FPs.** Each dot corresponds to one cell lineage. Colours as in Fig. 2. The number of cell lineages detected in each experiment is reported in the inset using two numbers. The one to the left is the number of data points shown in figure and the number to the right is the number of data points falling outside the axes' ranges. Each cell trap may contain more than one cell lineage. Only FPs where the number of detected cell lineages >5 for each experiment are shown.

outlier frequencies are higher than what is expected from the overall probability of incorrectly decoding a barcode. The other reason would be incorrect padlock hybridisation, but this type of error is unlikely when there are more than on RCA product per trap.

Finally, we grouped the fluorescence maturation curves based on which barcode was expressed. Solid lines in Fig. 5 show the average response for FPs from each of the three replicate experiments. The maturation curves display high reproducibility between the replicated experiments. As expected, we also observe that the slowly maturing proteins have a larger fold increase over time as they have a smaller fraction of mature fluorescent proteins when CHL is added.

We conclude that it is possible to read out single-cell barcodes expressed from the *E. coli* genome in a PDMS microfluidic device. This opens up for large-scale genomic engineering and phenotypic analysis with minimal interference from the barcoding system. This includes, for example, experiments based on mutations of promoters or protein-coding sequences of genes in their native chromosome position, fluorescence labelling of chromosomal loci for multiplexed chromosome tracking, and studies of how the position of a gene on the chromosome impacts its expression.

Under the experimental conditions used in this study, mCherry2-L and mCherry2 (see 'Methods' section for the minor difference between mCherry2-L and mCherry2) are the most rapidly maturing red FP and mScarlet-I is the brightest.

## Methods

### Statistics and reproducibility

Microscopy experiments and amplicone sequencing on the library of fp strains was performed in three replicate experiments, each starting from the cryostock of the library. Microscopy experiments underlying Supplementary Fig. 2 was carried out once for each +IPTG + CHL time-difference.

### Library construction

We selected 83 monomeric fluorescent proteins from the FPbase protein collection[8] with an excitation maximum between 559 nm and 685 nm. In addition, we included the mCherry-2L[7] variant, which lacks the potential multiple start codons of mCherry2. The amino acid sequences of each FP were back-translated and codon optimised in Benchling (https://benchling. com (2022) using *E. coli* as organism and medium GC content.

To produce barcodes, a set of ten thousand random DNA 30-mer sequences with an equal probability of all four bases were generated in silico, each with a GC content between 40% and 60% and with homopolimeric stretches <4 nucleotides long. Out of these, we selected sequences with a G or a C in the 5′ and 3′ end and with a Hamming distance >7 to every other sequence in the dataset, resulting in a list of 1202 sequences. A Hamming distance >7 ensures that the selected sequences are substantially different from one another, reducing the probability of spurious detection. From the resulting list, we arbitrarily picked 200 sequences, all containing a G in the beginning and the end and inserted them into the chromosome at *ygaY*

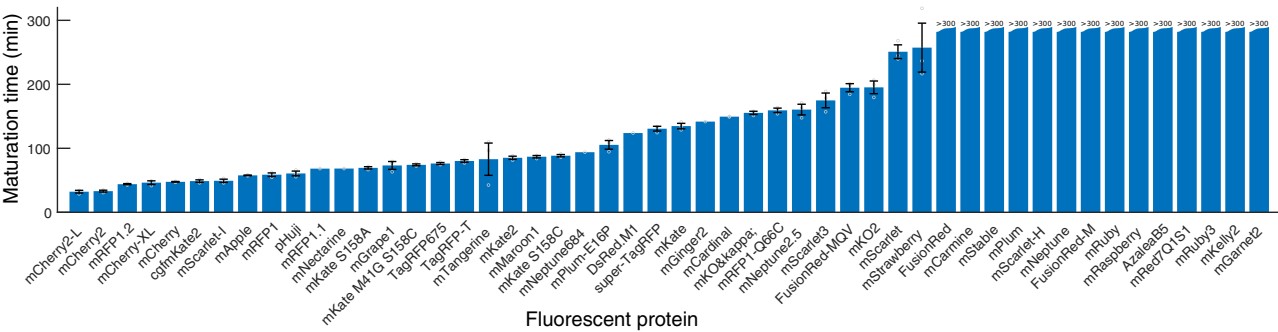

**Fig. 4 | Maturation times in replicate experiments.** Blue bars show the mean maturation time when more than one replicate maturation time is available, and the maturation time of a single replicate otherwise. Error bars show the standard error of the mean for the cases when three replicate estimates are available (see Supplementary Table 2 for individual maturation time estimates). Grey circles indicate the maturation time in each replicate experiment. The cases where the mean maturation time is longer than the fluorescence image acquisition time after +CHL are denoted '>300'. Only FPs with >4 decoded traps in an experiment are included in the mean and standard deviation calculation.

pseudogene with the same design as the above described library construction. Then we ran an on-chip genotyping experiment to screen for sequences that can be used as barcodes, i.e. that they are successfully detected using the on-chip genotyping experiment. Most barcodes used in the library were picked from this set of barcodes.

To enable chromosomal insertion of the selected fluorescent proteins, we designed gene fragments, each containing a gene for a fluorescent protein, a corresponding barcode, accessory sequences and homology sequences used for lambda red recombination. The terminator between the FP and the barcode is a strong synthetic terminator (L3S2P21) and consists of a short RNA hairpin followed by a U-rich sequence[16]. The barcode is under control of a T7 promoter, and included on the same RNA transcript are two accessory sequences: a spacer and the d0 sequence[17] that we have used previously because of its enhanced RNA stability[6]. As a control we also made a barcoded strain without a FP. For more details on the common motifs see Supplementary Table 1. Gene fragments were ordered as double-stranded DNA either from Twist Bioscience, IDT, or Eurofins. The ordered sequences are in Supplementary Data 1.

We used lambda red recombination[18] to insert a selectable/counterselectable cassette Atox1[19] into the *lac-operon* of an MG1655 *E. coli* strain (Supplementary Fig. 1). The inserted Atox1 cassette replaces most of the *lac*-operon, retaining only the promoter and the three operator sites (Supplementary Fig. 1). Successful chromosomal integration of the FP and barcode gene fragments using lambda red replaced the Atox1 cassette. The final desired scar-free constructs (Supplementary Fig. 1) were sequenced-verified using Sanger sequencing, with primer pairs, CCCATCTA-CACCAACGTGA and TTGTTCCTGCGCTTTGTTC. Both lambda red recombinations relied on expressions of lambda red genes, encoded on a plasmid, pSIM5-tet (with tetracycline resistance marker)[20]. For counterselection of Atox1 cassette, colonies were grown and selected on M9 plates containing Rhamnose. The pSIM5-tet plasmid was cured from the final strains by streaking single colonies and growing on LB agar plates at 42 °C over-night and the loss of pSIM5-tet was confirmed by streaking on LB agar plates with tetracycline (no growth). The 85 strains were grown overnight on a 96-well plate using 200 μl of LB media and on the next morning we pooled the strains together to create the redFP-library. The library was stored as a glycerol stock.

**Padlock probe design**
We designed PLPs against the 85 sequences generated above using a custom script that performs the following steps: each target sequence is reverse-complemented and split in half to produce 5′ and 3′ hybridisation arms. The scaffolds between the 2 arms are then filled with a set of custom sequences, each including a unique readout barcode for sequential hybridisation, a common sequence to prime the RCA reaction and other accessory

sequences that can be used, if necessary, to prime PCR reactions for in vitro analysis. At the end of this process, each padlock can target one of the 85 sequences above and can be decoded by a unique sequence of colours across multiple hybridisation cycles. PLPs for the 85 target sequences were pooled at 200 μM total concentration and phosphorylated (10 μl PLP pool, 2 μl T4 PNK (NEB M0201S), 5 μl ATP 10 mM, 5 μl PNK buffer 10×, 28 μl nuclease-free water; 30 min @ 37 °C, 20 min @ 65 °C), resulting in a pool at 0.46 μM per oligo. The list of PLPs used is found in Supplementary Data 2.

**Phenotyping**
In the microscopy experiments, the media used was M9-succinate (100 μM CaCl₂, 2 mM MgSO₄, 1X M9 salts and 0.4% (wt/vol) succinate (Sigma)), supplemented with 1X RPMI 1640 amino acid mix (Sigma), Uracil 0.1 mM (Sigma), 76.5 μg/ml Pluronic F108 (Sigma) and, unless otherwise stated, 1 mM IPTG.

Microscopy experiments were carried out on a Ti2-E microscope (Nikon) for both phase contrast and epifluorescence. The microscope was equipped with a 100× CFI Plan Apo Lambda DM (Nikon) objective, and the build-in intermediate magnification was set to 1.5× Images were acquired using a Sona camera (Andor).

In the *Maturation time measurements* experiments, epifluorescence imaging was carried out using a Spectra III (Lumencor) set to the Yellow channel and a filter cube consisting of a FF01-559/34 (Semrock) excitation filter, a T585lpxr (Chroma) dichroic mirror and a T590LP (Chroma) emission filter. In the *Measurement of the time to protein synthesis inhibition after + CHL* experiments, epifluorescence imaging was carried out using a Spectra III (Lumencor) set to the Teal channel and a filter cube consisting of a FF01-514/3 (Semrock) excitation filter, a Di02-R514 (Semrock) dichroic mirror and an ET550/50 M (Chroma) emission filter. To maintain a constant temperature of 30 °C, we used a temperature unit and lexan enclosure manufactured by Okolab for the microscope stage and the sample. The control of the microscope setup and the acquisition of images was done using micro-manager[21] running custom scripts.

In the genotyping experiments, the same microscope setup as for phenotyping was used except for the filter cubes and channel on the Spectra III. The light-source channel and filter combinations used were specific to each dye, as follows: DAPI: Violet channel, FF409-Di03 (Semrock), FF01-377/50 (Semrock), FF02-447/60 (Semrock); Alexa488: Cyan channel, FF-506-Di03 (Semrock), EX450-490 (Nikon), FF01-524/24 (Semrock); Cy3: Green channel, FF562-Di03(Semrock), FF01-543/22 (Semrock), FF01-586/20 (Semrock); Cy5: Red channel, FF660-Di02 (Semrock), 692/40 (Semrock); Alexa750: NIR channel, T760lpxr (Chroma), ET811/80 (Chroma).

The microfluidic chip was made of PDMS (SYLGARD 184) that was cured overnight (80 degrees) on a structured silicon wafer (ConScience), and the structured PDMS chip was bonded to a glass No. 1.5 coverslip

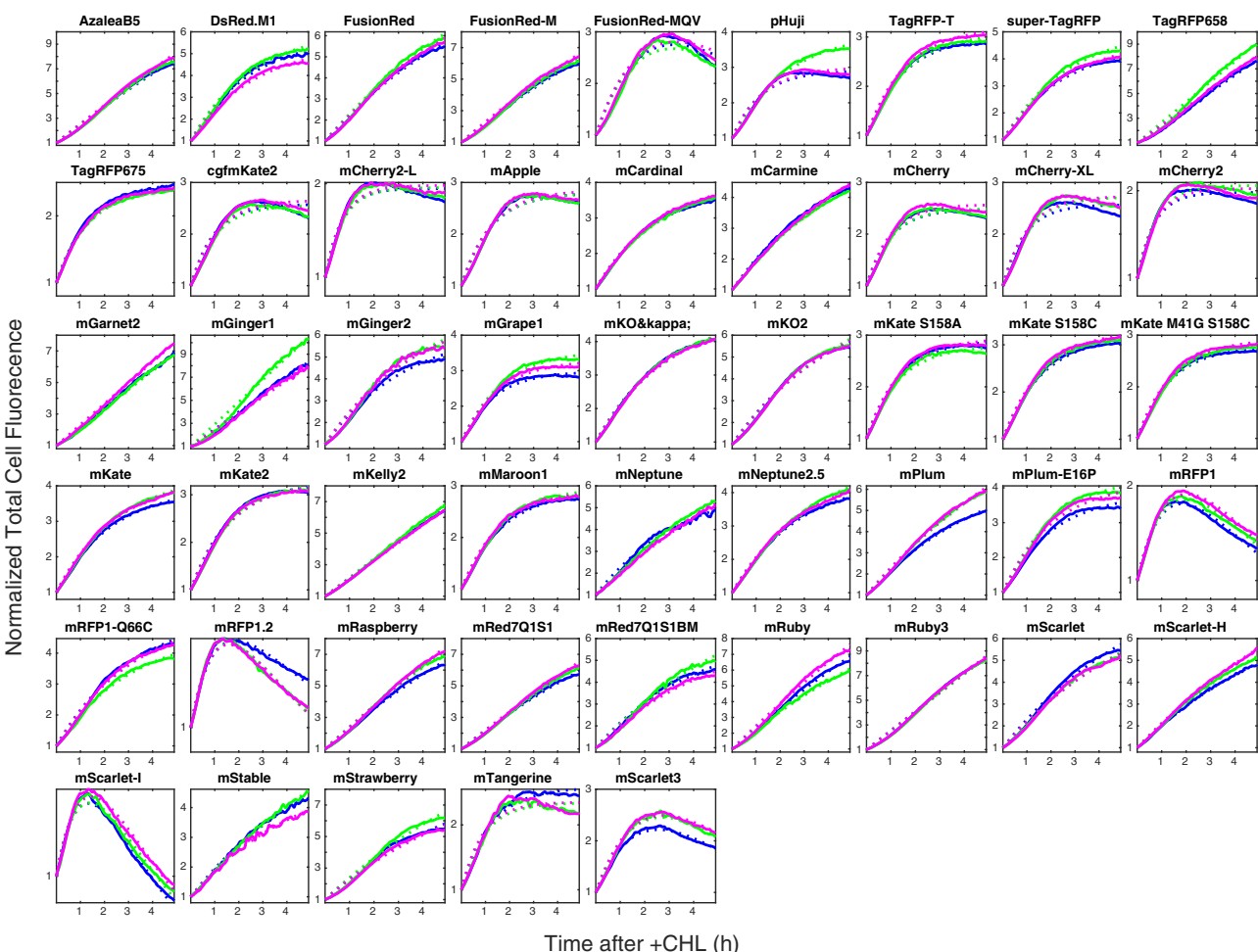

**Fig. 5 | Average total cell fluorescence as a function of time after + CHL for different FPs.** Colours as in Fig. 2. Only FPs where the number of detected cell lineages >5 for each experiment are shown. Solid lines are experimental data and dotted lines are fits to either a single or double exponential function (see 'Methods' for function definitions).

(Menzel-Gläser) as described in ref. 9. The chip was punched to allow for rapid switching of media as in ref. 22 and the media flows was controlled by an OB1 MkIII (Elveflow) electro pneumatic controller[22].

For growth of the cells before loading into the microfluidic chip, the cells were inoculated from cryostock and grown overnight in experiment media without IPTG at 37 °C in a shaking incubator. Next day, cells were diluted 1:250 in fresh media with 1 mM IPTG and grown for 4 h (maturation time measurements), or diluted in fresh media without IPTG and grown for 3 h (measurement of time to protein synthesis inhibition after +CHL). The cells were then loaded onto the mother machine chip.

For the maturation time measurements, the cell of the pooled library was, after being loaded into the microfluidic chip, grown in IPTG-containing media for 8 h to ensure that all cells in traps are direct descendants of the cell at the bottom of the trap. 100 positions were imaged for 6 h every minute in phase contrast (80 ms exposure time) and every 5 min in epifluorescence (100 ms exposure time). One hour after the imaging started the experiment media was swapped to experimental media containing chloramphenicol (250 µg/ml).

*For measurement of the time to protein synthesis inhibition after + CHL*, the strains EL3853[23], carrying an IPTG inducible SYFP2 (MG1655 ygaY::(AmpR paFAB120 Osym SYFP2)) and EL330 (MG1655, without any fluorescent protein) were grown as described above before loaded onto different sides of a mother machine chip. Cells were grown in the chip at 30 degrees for at least 3 h before image acquisitions were started. To allow cell tracking, the cells were imaged in phase contrast (80 ms exposure time) every second minute throughout the experiment. To estimate the 'leaky'

expression of SYFP2 without IPTG induction, cells were imaged in fluorescence (300 ms exposure time) prior to any media swaps. 1 mM IPTG and 250 µg/ml CHL were added with time differences indicated in Supplementary Fig. 2, where each time-difference corresponds to one experiment started from cryostock. Following the media swaps cells were again imaged in fluorescence 20 min after +CHL to allow for SYFP2 maturation.

**Genotyping**

After phenotyping, cells were fixed and permeabilized with 70% EtOH for 20 min followed by 20 min rehydration in PBS with 0.1% Tween (PBS-T) at RT.

From this stage all reaction mixes were administered to the front channel of the chip using a syringe pump and the lexan stage enclosure was kept at 30 degrees. The composition of the reaction mixes are given in Supplementary Table 4 and are administered as described below: (1) To further improve cell permeabilization we flowed the lysozyme reaction mix at 1 µl/min for 10 mins onto the chip; (2) Lysozyme activity was stopped by flowing the BSA solution mix at 1 µl/min for 10 min; (3) To produce the RNA barcodes used for PLP binding we flowed the zombie transcription mixture to the chip at a rate of 0.5 µl/min for 120 min; (4) To allow the padlock probe hybridisation to the corresponding RNA target we flowed the PLP hybridisation mixture to the chip at 1 µl/min for 30 min; (5) For the ligation of the padlocks hybridised to the target RNA, a SplintR ligase mixture was flown into the chip at 0.75 µl/min for 60 min; (6) To optimise the RCA reaction, the RCA primer hybridisation mixture containing a primer complementary to the padlock probe was flown into the chip at 1 µl/

min for 30 min; (7) For amplification of the padlock DNA we flowed the RCA reaction mixture onto the chip at 0.5 µl/min for 120 min. (8) Barcode detection: for each round of sequential FISH, the L-probes, which were pre-hybridised to detection oligos, were flown onto the chip at 1 µl/min for 30 min followed by imaging for genotyping; (9) Probes were removed from the RCA product by flowing the probe stripping mixture at 1 µl/min for 30 min. Steps 8 and 9 were repeated an additional 6 times with different known variations of L-probe detection oligo pools.

The L-probes were mixed for each hybridisation-round according to the code allowing for error correction (See Supplementary Data 3 for a list of the L-probes used). Each strain was assigned a randomly chosen 7-symbol code word (with an alphabet of 4 symbols), while ensuring that each code word has a Hamming distance of at least 3 symbols from all other 84 code words. Each symbol in the code is represented by a differently coloured detection oligo. Thus, for each hybridisation-round we mixed a collection of 85 L-probes each connecting the PLPs readout barcode and the correct detection oligo for that round.

After all genotyping rounds, DAPI stain at 1 µg/ml (thermo scientific) was flown into the chip for 30 min which was then imaged using the DAPI optical setting.

### Amplicon sequencing of barcodes

3 ml culture was harvested directly from the same cultures used for maturation time estimates and stored at −20 °C. Genomic DNA was extracted using DNeasy UltraClean Microbial Kit (Qiagen #12224-50) and stored at 4 °C ON. Phusion PCR was performed using the following primers: acactctttccctacacgacgctcttccgatctCTCGGTACCCAAATTCCAGAAAAG, and gtgactggagttcagacgtgtgctcttccgatctCTCTTGTCTCCTTGCGCTAG, and PCR products were purified using Monarch PCR clean up kit. Index primers were added using Phusion PCR. Samples were sequenced using an iSeq (Illumina) following the machine manual. The sequencing reads were sorted according to exact matches with each of the different barcodes, and the reads for each barcode were counted.

### Image analysis

Image analysis was done using an in-house pipeline. In the pipeline, cells are first segmented using a U-Net convolutional network based approach[24]. The segmented cells are then tracked from frame to frame and linked into lineages using the Baxter algorithm[25]. Total cell fluorescence was calculated by summing up camera pixel intensities inside the segmented cell outline. To only include pixel intensity due to fluorescence, the average pixel intensity of an area without either cells or PDMS bonded directly to glass, was subtracted from each pixel value before making the sum in each cell. Cells without an FP showed no discernible signal. Given the high fluorescence intensity variability between different FPs and the small spatial distance between adjacent cell traps in the microfluidic chip, the signal from cells carrying a high intensity FP may overshadow the signal of cells in neighbouring traps which carry a low intensity FP. To overcome the problem with signal bleeding between cell traps, we only included cells where the intensity density inside the segmented cells is, on average, 3-fold higher as compared to the cell's immediate surroundings. The tracked cell lineages were used to generate one maturation curve for each entire lineage where the areas and total fluorescence of two daughters, four granddaughters, etc, were summed up for each time point. Only cell lineages that existed at the time of swapping to chloramphenicol-containing medium and that were possible to track until the end of the experiment were used in calculating the maturation times.

### Maturation time calculations

To estimate the FP maturation time, the total cell fluorescence after chloramphenicol treatment is fitted to either a function consisting of a single exponential term, or a function consisting of the sum of two exponential terms. These model-fits assume that the maturation process is well described by a single step reaction and that protein production is instantly, and completely, stopped at the time of exposing the cells to chloramphenicol. The latter assumption was tested separately

(Supplementary Fig. 2), where we found that expression of FP completely stops within a few minutes.

In cases where the total fluorescence is reaching a constant plateau, it is fitted to the function

$$F(t) = c(\alpha + 1 - \alpha \exp(-t/\tau_m))$$

where $F$ is the total cell fluorescence, $t$ is time after chloramphenicol addition, $\tau_m$ is the FP maturation time, and $c$ and $\alpha$ are fitting constants.

In cases where the total fluorescence is clearly decreasing at the end of the experiment such that the ratio between the peak fluorescence and the end point fluorescence, on average for all cells in one experiment carrying the same FP, is less than 0.85, the total fluorescence after chloramphenicol addition is fitted to

$$F(t) = c[(\tau_b/(\tau_b - \tau_m))(\alpha + 1)\exp(-t/\tau_b) - \alpha(\tau_b/(\tau_b - \tau_m))\exp(-t/\tau_m)]$$

where $\tau_b$ is the characteristic time for signal decrease due to bleaching.

### Phenotypic outlier detection

To quantify the number of cell traps which are showing a phenotype that is deviating from the behaviour of the majority of cells in traps genotyped as containing the same FP, we used a clustering based analysis of the data shown in Fig. 3. First the data from the three different repeat experiments were pooled. After this, we used DBSCAN[26] (built-in function in MATLAB R2022b, MathWorks) to find clusters. The cluster containing the largest number of cells was identified as the major cluster. Finally, the number of traps in the major cluster and the number of traps outside the major cluster were counted. Only traps where the number of identified lineages were more than 2, and where all the cells in the trap were unanimously identified as either being inside or outside the major cluster were used. Only FPs where the number of traps in the major cluster where more than 5 were included. The fraction of traps with deviating phenotypes are shown in Supplementary Table 3A.

### Detection of barcodes

Phase contrast images from each round of genotyping were aligned using the dot unique position identifiers imprinted next to empty traps. Trap segmentation was performed using the deep learning model described in ref. 27. Non-empty traps were detected by identifying traps that include permeabilized cells using DAPI staining. Fluorescent blobs were detected using a signal threshold that was five standard deviations higher than the mean of the Gaussian fitted to the histogram of pixel intensities from all of the traps in each position in the same fluorescence channel. The resulting fluorescence masks were compared between rounds and the final mask consisted of the pixels that appeared in at least half of the rounds. For trap code assignment, the fluorescent signal from a trap was considered only if (i) it passed the intensity threshold, (ii) the number of pixels passing the intensity threshold passed an area threshold, and (iii) the signal was within the accumulated fluorescence mask. If after applying the thresholds and the mask there was a signal in a single channel, it was designated as the decoded channel. Otherwise, if there was a signal in multiple channels, a channel was designated as the decoded channel only if its total intensity was at least twice higher than all other channels. We applied error correction to trap codes fixing at most 1 round, by assigning the decoded sequence to one of the code words if it was identical to this code word, or one error away from it (i.e. Hamming distance equals 1). We assess single round error probability by counting all the rounds with a wrong signal based on error correction, divided by the total number of rounds in valid traps.

### Reporting summary

Further information on research design is available in the Nature Portfolio Reporting Summary linked to this article.

## Data availability

Microscopy images and the corresponding output from the image analysis pipeline are found at the Bio Image Archive, https://doi.org/10.6019/S-BIAD1353. Output of amplicon sequencing is found at the European Nucleotide Archive (ENA) at EMBL-EBI under accession number PRJEB89507.

## Material availability

Strains used in this study are available on reasonable request.

## Code availability

The code for image analysis of microscopy images is found at the Bio Image Archive, https://doi.org/10.6019/S-BIAD1353. The code for post-processing of image-analysis output and for generating all tables and figures is found at the SciLifeLab Data Repository, https://doi.org/10.17044/scilifelab.26976952.

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

## Acknowledgements

We are grateful for the helpful contributions from Vinodh Kandavalli, Daniel Camsund, Nick Shakari, Nicole Eger and Irmeli Barkefors. This study was made possible by grants from the ERC (advanced grant no. 885360), the Swedish Research Council (grant nos. 2016-06213 and 2018-03958 to J.E., as well as 2019-01238 to M.N.), the Knut and Alice Wallenberg Foundation (grant nos. 2016.0077, 2017.0291 and 2019.0439), and the eSSENCE e-science initiative. The computations and data management were enabled by resources provided by the Swedish National Infrastructure for Computing at UPPMAX, partially funded by the Swedish Research Council through grant agreement no. 2018-05973.

## Author contributions

A.K., D.A.G.S and B.C.S. designed and constructed strains; R.R.G.S., J.L., M.G., M.N., F.S and J.E. developed the genotyping protocol; M.R. designed barcode sequences; J.L. made the maturation time and protein synthesis inhibition experiments; B.C.S. made the amplicon sequencing experiments; D.S. and S.Z. developed the barcoding decoding method; D.S. analysed the in situ genotyping data. D.A.G.S. and D.F. analysed the phenotyping data; J.E. and D.F. conceived the project and wrote the manuscript with input from all authors.

## Funding

## Competing interests

The authors declare the following competing interests: J.E. patented optical pooled screening in 2014 (WO2016/007063 A1) and holds shares in Bifrost Biosystems. M.N. holds shares in Bifrost Biosystems.
