## [Transparent Peer Review file · Communications Biology]

Pooled optical screening in bacteria using chromosomally expressed barcodes

Corresponding Author: Professor Johan Elf

This manuscript has been previously submitted to another journal. This document only contains information relating to versions considered at Communications Biology.

Version 0:

Reviewer comments:

Reviewer #1

(Remarks to the Author)
SUMMARY

This paper discusses a method for in situ, single cell genotyping of a pooled mutant library using chromosomal T7 expressed barcodes. It uses a seqFISH like approach to achieve this, and it is based on significant past work by the same group. In this paper, they demonstrate that they can accurately detect barcodes from a set of strains encoding various fluorescent proteins at the lacZ genomic locus. They find that the relative frequencies of the strains correlate well between a bulk amplicon sequencing experiment and their single cell microscopy. The paper shows that maturation curves could be generated for each of the FP variants and replicate experiments show very similar results, suggesting that the method has high fidelity. Finally, they assess the single cell data for outliers and find some, but overall see data cluster tightly, indicating high fidelity genotyping and measurement.

OVERALL IMPRESSION

This is a very short paper that describes a high-quality experiment and convincingly demonstrates in situ genotyping of chromosomally expressed barcodes, as the authors claim. Therefore, I believe this paper should be accepted with revisions. However, I feel that the paper could be made clearer with changes to the text and figures (particularly Figs 2 and 3). Also, it remains a bit unclear how much of the seqFISH style genotyping is the same as their previous work. They should add further details about the barcode decoding scheme and the actual genomic locus they used. Ultimately, the data in this paper support the author's claims and I think this is an interesting contribution to the literature that will be valuable. This work may open interesting avenues for pooled mutant library screens with microscopy readouts.

SPECIFIC COMMENTS

Major comments:

1. Since the primary claim is about chromosomally expressed barcodes, there should be more information in main figures and text about this. The following questions and comments should be addressed.

- How similar is the seqFISH style genotyping to previous work by the same group with plasmid libraries?
- Why would it be more difficult to decode genomic barcodes?
- Figure 1 should show the chromosomally expressed barcode locus (i.e. a better version of Fig. S1).
- This reviewer is familiar with seqFISH, but most readers of this paper probably will not be. I strongly recommend adding to Fig. 1 or at least a supp figure showing the actual barcoding scheme and how different probe sets / color channels can be used to decode.

2. Figures 2 and 3 are not very informative. For both figures, I recommend showing several representative FP plots, but moving most of the panels to the supplement. Then create summary plots with derived metrics to show the whole dataset, for example, show the maturation times instead of every single maturation curve. Here are further specific suggestions that may be helpful:

- For Figure 2, why not show a plot with median maturation times and number of lineages for each FP. The data from Table S2 would be more helpful as a plot and would be a more appropriate representation than showing all FP maturation curves.

- The number of lineages from Table S2 should certainly be in a main figure. Why does the number of lineages vary so much (>10x). Wouldn't you expect the strains to be quite evenly distributed? Or is this the expected distribution from loading single cells? The authors should comment on this. Or does this mean certain barcodes were difficult to decode?

- Show the NGS comparison in a main figure. Specifically, a scatterplot with the correlation between NGS frequency and in situ frequency. Or show the log₂ in situ / NGS from S4C – this could also just be shown as boxplot to make it clear the proportion that fall within the 2x range mentioned in the main text.

- Should the outlier analysis in figure 3 come before figure 2? This would be the logic: Fig 1 is the approach. Fig 2 could show that the barcodes can be accurately detected (# lineages for each FP and outlier analysis). Fig. 3 shows the final result – highly reproducible maturation curves and some insights into the FPs.

Minor comments:

1. Regarding Fig. S1. Are the FPs replacing the lacZ gene or part of the lacZ?

- What does 'in-frame of the lacZ gene' mean? An in frame knockout would be deleting everything between the start and stop codon. An in frame fusion would be creating a translational fusion. Based on Fig. S1 neither of these apply, so it is quite unclear.

- Based on the proximity of cynX in Fig. S1, do the authors mean the FP construct replaces the lacY and lacA genes? If so, this should be stated much more clearly.

2. Fig. S1 doesn't show the counter selection scheme used.

3. It should be more clearly stated in the methods that FPs and barcodes with flanking homology arms were ordered as gene fragments and used directly as double stranded DNA substrates for lambda red recombination. If this was not how the genetics was done, then more detail should be added to adequately explain. I also recommend adding a short explanation of the lambda red protocol use - which helper plasmid, how was helper removed, how was atox selected against etc.

4. Upon initial reading of the main text, it is unclear what CB5644 type identifiers mean. I see in the figure legends that they are likely replicate experiments, but it should be clearly stated in the main text.

5. It may be worth further discussing the types of chromosomally encoded pooled mutant libraries that might benefit from this type of genotyping / microscopy screen.

6. "This has not been possible in bacteria" statement from abstract is strange to me. It implies that expressing barcodes off the genome was not possible. Instead, I would suggest that expressing and genotyping barcodes off the genome had not been "achieved" in bacteria.

7. Fig S4 has an issue with div tags in the plot labels. For example <in situ>

8. The authors should consider discussing how their barcoding scheme scales to mutant libraries of different sizes.

Reviewer #2

(Remarks to the Author)

Major

1) The explanation is insufficient, and the experiment and materials are complex, making it difficult to understand.

- While it is stated that each trap in the microfluidic chip consists of a single chromosomal barcode after pooling, the process of how this is achieved is not described in detail.

- Although it is mentioned that data is collected on a trap basis during the phenotyping and genotyping processes, the detailed procedures for each process are lacking.

- The relationship between the column, trap, and barcode in the experiment's setup and interpretation is unclear.

Minor

1) Detailed information on the tools used for chromosomal insertion is lacking, such as the specific plasmids (vectors) or tools (e.g., the lambda red recombination system).

2) The description of the genetic engineering procedures for inserting barcodes and fluorescent proteins in the MG1655 strain is insufficient.

- 3) The explanations for abbreviations like BC, PDMS, and CHL are lacking.
- 4) The meaning of "zombie transcription" in Figure 1 is difficult to interpret.

Version 1:

Reviewer comments:

Reviewer #1

(Remarks to the Author)

The authors have done a good job of addressing my original comments. The work supports the authors' claims and it will be interesting for a wide range of researchers. I recommend that the journal publishes this manuscript.

I found two very minor typos:

- dependent "which suggests a barcode sequence dependant variability in the barcode readout efficiency."
- barcode "and no FP control strain baccodes (Table S3B)."

Reviewer #2

(Remarks to the Author)

I confirmed that all the major and minor issues I previously raised have been resolved by adding examples/figures and providing more detailed descriptions of the methods. The authors' additional revisions have clarified the meaning, and I believe this manuscript now meets the qualifications for publication in this journal. Thank you.

Reviewers' comments:

Reviewer #1 (Remarks to the Author):

SUMMARY

This paper discusses a method for in situ, single cell genotyping of a pooled mutant library using chromosomal T7 expressed barcodes. It uses a seqFISH like approach to achieve this, and it is based on significant past work by the same group. In this paper, they demonstrate that they can accurately detect barcodes from a set of strains encoding various fluorescent proteins at the lacZ genomic locus. They find that the relative frequencies of the strains correlate well between a bulk amplicon sequencing experiment and their single cell microscopy. The paper shows that maturation curves could be generated for each of the FP variants and replicate experiments show very similar results, suggesting that the method has high fidelity. Finally, they assess the single cell data for outliers and find some, but overall see data cluster tightly, indicating high fidelity genotyping and measurement.

OVERALL IMPRESSION

This is a very short paper that describes a high-quality experiment and convincingly demonstrates in situ genotyping of chromosomally expressed barcodes, as the authors claim. Therefore, I believe this paper should be accepted with revisions.

We are grateful for these appreciative comments.

However, I feel that the paper could be made clearer with changes to the text and figures (particularly Figs 2 and 3). Also, it remains a bit unclear how much of the seqFISH style genotyping is the same as their previous work. They should add further details about the barcode decoding scheme and the actual genomic locus they used.

Please see specific comments below

Ultimately, the data in this paper support the author's claims and I think this is an interesting contribution to the literature that will be valuable. This work may open interesting avenues for pooled mutant library screens with microscopy readouts.

SPECIFIC COMMENTS

Major comments:

1. Since the primary claim is about chromosomally expressed barcodes, there should be more information in main figures and text about this. The following questions and comments should be addressed.

- How similar is the seqFISH style genotyping to previous work by the same group with plasmid libraries?

In the previous work, the barcode RNA was expressed from plasmid by inducing T7 expression before fixing the cells. The RNA was probed with one FISH probe per barcode such that each barcode only gave fluorescence in one round of probing. The number of identified strains is therefore \$C \times N\$, where C is the number of colors and N is the number of rounds of probing.

In the new protocol, the barcode RNA is expressed from the chromosome by T7 added to the fixed and permeabilized cells, which gives more expression per locus. The barcode signal is further amplified by padlock ligation and rolling circle amplification. This allows for combinatorial barcode readout, where each barcode gets a color in each round. The maximal number of identified strains are therefore \$C^N\$, where C is the number of colors and N is the number of rounds of probing.

This is now better described in the main text of the manuscript.

- Why would it be more difficult to decode genomic barcodes?

It is more difficult since fewer barcode RNAs are expressed and it requires amplification of the barcode signal to be able to perform combinatorial FISH.

- Figure 1 should show the chromosomally expressed barcode locus (i.e. a better version of Fig. S1).

Figure 1 now includes an illustration of what has been introduced and where in the chromosome.

- This reviewer is familiar with seqFISH, but most readers of this paper probably will not be. I strongly recommend adding to Fig. 1 or at least a supp figure showing the actual barcoding scheme and how different probe sets / color channels can be used to decode.

Figures 1C and D have been reworked to emphasize how the actual barcoding scheme works.

2. Figures 2 and 3 are not very informative. For both figures, I recommend showing several representative FP plots, but moving most of the panels to the supplement. Then create summary plots with derived metrics to show the whole dataset, for example, show the maturation times instead of every single maturation curve.

We have added a bar plot of the maturation times, which admittedly give more direct access to the whole dataset. However, since the focus is on the method for combining live cell phenotyping and genotyping, we would also like to keep the time courses and distributions in the main text since they better describe the reproducibility of the method (Figs 2 and 3), the accuracy of barcode readout (Fig 3), and the time-dependent aspect of the phenotyping (Fig2).

Here are further specific suggestions that may be helpful:

- For Figure 2, why not show a plot with median maturation times and number of lineages for each FP. The data from Table S2 would be more helpful as a plot and would be a more appropriate representation than showing all FP maturation curves.

The barplot with median maturation times has been included as Fig 4. The number of lineages is a direct consequence of the in situ detected frequency of genotypes in the library. Please see below.

- The number of lineages from Table S2 should certainly be in a main figure. Why does the number of lineages vary so much (>10x). Wouldn't you expect the strains to be quite evenly distributed? Or is this the expected distribution from loading single cells? The authors should comment on this. Or does this mean certain barcodes were difficult to decode?

The requested information is in Fig S4, which now has been moved into the main text as Fig. 2. Panel A shows that the library has a relatively balanced frequency of each genotype. B shows the frequency of strains detected in situ. C shows the mismatch between A and B, where 10 of the strains are significantly underrepresented (in situ frequency / library frequency <0.5) and 2 strains are significantly overrepresented (in situ frequency / library frequency >2). We do not expect this to be because of the loading of single cells, but because of differences in barcode readout efficiency. This is now clarified in the main text.

- Show the NGS comparison in a main figure. Specifically, a scatterplot with the correlation between NGS frequency and in situ frequency. Or show the log2 in situ / NGS from S4C – this could also just be shown as boxplot to make it clear the proportion that fall within the 2x range mentioned in the main text.

Fig S4 is now moved to the main text. Please see the previous comment.

- Should the outlier analysis in figure 3 come before figure 2? This would be the logic: Fig 1 is the approach. Fig 2 could show that the barcodes can be accurately detected (# lineages for each FP and outlier analysis). Fig. 3 shows the final result – highly reproducible maturation curves and some insights into the FPs.

We also think this is a good idea and have restructured the text accordingly.

Minor comments:

1. Regarding Fig. S1. Are the FPs replacing the lacZ gene or part of the lacZ?

This has been clarified in the revised Fig S1.

- What does 'in-frame of the lacZ gene' mean? An in frame knockout would be deleting everything between the start and stop codon. An in frame fusion would be creating a translational fusion. Based on Fig. S1 neither of these apply, so it is quite unclear.

Both the methods section and Fig. S1 have been updated to clarify the details of the strain construction.

- Based on the proximity of cynX in Fig. S1, do the authors mean the FP construct replaces the lacY and lacA genes? If so, this should be stated much more clearly.

That was indeed the case. This fact has been more clearly indicated in Fig. 1, in Fig. S1 and in the methods section.

2. Fig. S1 doesn't show the counter selection scheme used.

The previous figure S1 was misleading and is now clarified in Fig1. The cloning steps, and details of the constructs, are now described in the new Fig S1.

3. It should be more clearly stated in the methods that FPs and barcodes with flanking homology arms were ordered as gene fragments and used directly as double stranded DNA substrates for lambda red recombination. If this was not how the genetics was done, then more detail should be added to adequately explain. I also recommend adding a short explanation of the lambda red protocol use - which helper plasmid, how was helper removed, how was atox selected against etc.

We have added an extended description to the methods sections and improved Fig. S1 to include the lambda-red steps.

4. Upon initial reading of the main text, it is unclear what CB5644 type identifiers mean. I see in the figure legends that they are likely replicate experiments, but it should be clearly stated in the main text.

This has now been clarified in connection to Table 1.

5. It may be worth further discussing the types of chromosomally encoded pooled mutant libraries that might benefit from this type of genotyping / microscopy screen.

A list of examples was in the introduction of the previous version. This section is now highlighted in green. We have also expanded the last paragraph of the main text to include more examples of the benefits of the method.

6. "This has not been possible in bacteria" statement from abstract is strange to me. It implies that expressing barcodes off the genome was not possible. Instead, I would suggest that expressing and genotyping barcodes off the genome had not been "achieved" in bacteria.

Good point!

7. Fig S4 has an issue with div tags in the plot labels. For example

We hope that the new main text version of the figure has these issues resolved.

8. The authors should consider discussing how their barcoding scheme scales to mutant libraries of different sizes.

The C^N scaling of combinatorial barcode identification has now been explicitly written out.

Reviewer #2 (Remarks to the Author):

Major

1) The explanation is insufficient, and the experiment and materials are complex, making it difficult to understand.

A number of points have been clarified in response to the comments of reviewer one. We hope that the new version is more accessible.

- While it is stated that each trap in the microfluidic chip consists of a single chromosomal barcode after pooling, the process of how this is achieved is not described in detail.

It is now described in detail how growth and division of the mother cell in the bottom of the cell trap pushes out cells from other lineages.

- Although it is mentioned that data is collected on a trap basis during the phenotyping and genotyping processes, the detailed procedures for each process are lacking.

In the revised version we now clarify that the phenotypes are mapped to the genotypes at the single trap level using Unique Position Identifiers imprinted in the microfluidic device.

- The relationship between the column, trap, and barcode in the experiment's setup and interpretation is unclear.

We mistakenly used the word barcode both for (i) the RNA based genetic barcode used for strain identification and for (ii) the spatial barcode imprinted in the microfluidic chip and used for alignment between the phenotyping and genotyping rounds. The latter type of barcode is now renamed "unique position identifier" to avoid confusion. We apologize for the confusion.

Minor

1) Detailed information on the tools used for chromosomal insertion is lacking, such as the specific plasmids (vectors) or tools (e.g., the lambda red recombination system).

We have extended the description of the strain construction as was also requested by reviewer 1.

2) The description of the genetic engineering procedures for inserting barcodes and fluorescent proteins in the MG1655 strain is insufficient.

We have extended the description of the strain construction as was also requested by reviewer 1.

3) The explanations for abbreviations like BC, PDMS, and CHL are lacking.

This has been fixed in the revised version.

4) The meaning of "zombie transcription" in Figure 1 is difficult to interpret. This is now explained in the main text.